YH-RTYO: an end-to-end object detection method for crop growth anomaly detection in UAV scenarios

Li Yihang 1
Yang WenZhong yangwenzhong@xju.edu.cn 2
Lu Zhifeng 3
Shi Houwang 1
1 College of Computer Science and Technology, Xinjiang University , Urumchi , China
2 Xinjiang Key Laboratory of Multilingual Information Technology, Xinjiang University , Urumqi , Xinjiang Uygur Autonomous Regions , China
3 School of Information Science and Technology, Xinjiang Teacher’s College , Urumchi , China
Coelho Paulo Jorge
Electronic publication date: 2024 Dec 16
Publication date: 2024
Volume: 10
Electronic Location ID: e2477
Received 2024 Aug 6; Accepted 2024 Oct 14
Copyright: ©2024 Li et al.
Copyright year: 2024
Copyright holder: Li et al.
License: This is an open access article distributed under the terms of the Creative Commons Attribution License, which permits unrestricted use, distribution, reproduction and adaptation in any medium and for any purpose provided that it is properly attributed. For attribution, the original author(s), title, publication source (PeerJ Computer Science) and either DOI or URL of the article must be cited.
License URL: https://creativecommons.org/licenses/by/4.0/

Keywords: Growth conditions, Target detection, Convolutional model, Transformer, UAV scenes

Funding: The National Key Research and Development Program of China 2022ZD0115802 The Key Research and Development Program of the Autonomous Region 2022B01008 The National Natural Science Foundation of China 62262065 The Tianshan Science and Technology Innovation Leading talent Project of the Autonomous Region 2022TSYCLJ0037 This research was funded by the National Key Research and Development Program of China (Grant No. 2022ZD0115802), the Key Research and Development Program of the Autonomous Region (Grant No. 2022B01008), the National Natural Science Foundation of China (Grant No. 62262065), the Tianshan Science and Technology Innovation Leading talent Project of the Autonomous Region (Grant No. 2022TSYCLJ0037). The funders had no role in study design, data collection and analysis, decision to publish, or preparation of the manuscript.

==============================
Background

Small object detection via unmanned Aerial vehicle (UAV) is crucial for smart agriculture, enhancing yield and efficiency.

Methods

This study addresses the issue of missed detections in crowded environments by developing an efficient algorithm tailored for precise, real-time small object detection. The proposed Yield Health Robust Transformer-YOLO (YH-RTYO) model incorporates several key innovations to advance conventional convolutional models. The model features an efficient convolutional expansion module that captures additional feature information through extended branches while maintaining parameter efficiency by consolidating features into a single convolution during validation. It also includes a local feature pyramid module designed to suppress background interference during feature interaction. Furthermore, the loss function is optimized to accommodate various object scales in different scenes by adjusting the regression box size and incorporating angle factors. These enhancements collectively contribute to improved detection performance and address the limitations of traditional methods.

Result

Compared to YOLOv8-L, the YH-RTYO model achieves superior performance in all key accuracy metrics, with a 13% reduction in the scale of model. Experimental results demonstrate that the YH-RTYO model outperforms others in key detection metrics. The model reduces the number of parameters by 13%, facilitating deployment while maintaining accuracy. On the OilPalmUAV dataset, it achieves a 3.97% improvement in average precision (AP). Additionally, the model shows strong generalization on the RFRB dataset, with AP50 and AP values exceeding those of the YOLOv8 baseline by 3.8% and 2.7%, respectively.

Introduction

In the current global context of peace and economic development, increasing investments are being directed towards the service industry, the Internet, and other high-tech sectors. As traditional agriculture undergoes industrial restructuring and social development needs evolve, modern agriculture has become critically important. Modern agriculture is characterized by precision and intelligent farming, which are pivotal for advancing agricultural practices. Leveraging computer technology to enhance agricultural efficiency is crucial, as it plays a significant role in driving social progress. Unmanned aerial vehicle (UAV) remote sensing technology offers substantial potential for rapid and widespread assessment of crop growth due to its mobility, flexibility, broad coverage, and capacity to collect extensive data samples. With factors such as population growth and climate change intensifying pressures on food production, accurately monitoring crop growth is vital for ensuring food security and promoting sustainable development. Timely information on crop growth, nutrient status, and vegetation cover can lead to improved adjustments in irrigation, fertilization, and pest control measures, thereby increasing yields and minimizing environmental impacts. Accurate crop detection is essential for monitoring growth, health status, and pest and weed presence, which enhances planting strategies and yields while reducing labor and resource consumption.

Recent advancements in detection technologies include improvements in models such as the faster region-convolutional neural network (Faster R-CNN) for precise disease detection in tomatoes (Zhang, Song & Zhang, 2020), and UAV-based weed detection methods utilizing YOLOv7, which enhance detection in complex scenarios by employing specialized datasets (Gallo et al., 2023). Similarly, complex crop row detection algorithms have been proposed to achieve precise localization of crop rows using improved YOLO-R models (Ruan, Chang & Cui, 2023). Despite these advancements, existing research predominantly focuses on either broad macro-level studies or localized experimental research, often overlooking the detailed observation of individual plant growth statuses.

The rapid development of drone technology has expanded its applications across various fields, offering flexibility, efficiency, and versatility. In agriculture, drones equipped with high-resolution cameras and sensors provide precise images and data for crop monitoring and management. They facilitate automatic identification and detection of pests, diseases, weeds, and plant growth status, aiding farmers in making timely decisions to enhance crop yield and quality. However, UAVs capture farmland scenes from varying heights, leading to significant changes in target scales within images, with numerous small targets densely packed, resulting in pixelation and susceptibility to image noise. Objects smaller than 32 × 32 pixels are considered small, which poses challenges for network optimization and target localization.

YOLOv8, a one-stage convolutional network model known for its rapid training and reliability, faces limitations due to the convolutional layers’ inability to capture broader contextual information, excelling in local detection but struggling with larger areas. Additionally, the stacking of convolutional layers increases the model’s complexity. Conversely, Real-Time DEtection TRansformer (RT-DETR), a Transformer-based model within the DETR family, is renowned for its accuracy over large ranges and abundant samples, with fewer parameters. However, it suffers from longer training times and limitations in small object detection and scenarios with limited sample availability.

This study presents a fusion model combining RT-DETR and YOLOv8 to address issues related to small target pixel areas, limited features, noise interference, and low detection accuracy in UAV crop detection scenarios. We develop the RTYO model and further enhance it to create the Yield Health Robust Transformer-YOLO (YH-RTYO) model. The key innovations of this paper are as follows:

• Multi-scale fusion and model optimization: UAV-based small object detection in crops often faces challenges like missed detections in crowded scenes with overlapping targets, as current mainstream detectors are mainly designed for larger pixel-sized objects.Recognizing the complex and diverse scale variations in UAV crop growth datasets, this study leveraged the fast training speed and high precision of YOLO models for small targets, along with DETR’s proficiency in constructing large semantic relationship networks in scenarios with abundant targets and its fast inference speed. This combination led to the design of the RTYO model, where this research optimized its architecture to balance parameter count and computational efficiency.

• Diverse branch block technique of C2f: To improve feature extraction and generalization while managing training costs, this study design and implement the CSPDarknet53 to 2-Stage FPN with Diverse Branch Block (CFB) module. This approach enriches the feature space by incorporating branches of varying scales and complexities, addressing missed target detection issues, and enhancing recall rates and mean average precision (AP).

• Enhanced multi-scale feature fusion: In the local feature pyramid network (LFPN) module, it address the inefficiencies of repeated upsampling operations and simple 1 ×1 convolutions by implementing a more efficient CARAFE upsampling approach and FSM-enhanced branching, reducing feature redundancy and improving target boundary prediction accuracy.

• Advanced loss function design: To tackle unbalanced class distribution and large-scale variation, this study integrate factors from Inner-IoU, MPDIoU, and CIoU, adding a power factor θ to enhance the loss function’s penalty term.

Related Work

Convolutional neural network

The algorithmic models for the detection task are categorized into convolutional neural network detectors and Transformer-based detectors. Convolutional networks are good at learning at a small range of features, and as the network is designed deeper, the field of view of the convolution is wider and the area that can be learned is larger. The main algorithmic idea of the two-stage detection model is divided into two steps, first learning feature information to generate a candidate frame for detection, and then in the candidate region for more specific localization tasks and classification tasks target detection model of the famous region-based convolutional neural network (R-CNN), which is the pioneering work of introducing convolutional neural networks into the field of target detection (Girshick et al., 2014), Fast RCNN is a fast update of RCNN (Girshick, 2015). Faster RCNN is the set of RCNN, the first step generates the target box, and the second part then classifies and recognizes the targets inside the box, its accuracy and speed have a great advantage inside the RCNN family (Ren, 2015). Mask RCNN, compared to Faster RCNN, has a high accuracy but slightly complex mode (He et al., 2017). Cascade RCNN is an improvement of Mask RCNN in terms of the problem of setting queue values (Cai & Vasconcelos, 2018). These two-stage detectors have better capabilities and structural designs. Although these detectors achieve high accuracy, their slower speed makes them less practical for real-time applications compared to single-stage detectors.

The YOLO series of detection algorithm models detect the target directly from the image and do not need to be divided into two stages of training steps, which greatly reduces the time cost of training and inference (Redmon, Divvala & Girshick, 2016). The SSD model as well as its derivatives are also representative of the one-stage models, which have high accuracy and can also maintain the real-time detection speed of the detection model, but have difficulties in small target detection (Liu, Anguelov & Erhan, 2016). Real-time target detection has been dominated by YOLO series models in the past, which have been widely used in crop production, disaster prevention, yield estimation, and other major agricultural intelligent industries, where detecting and categorizing crop growth, weed control, pest control, and disease prevention are common and important tasks, which provide more accurate information about the agricultural situation, and assist the intelligent system in realizing the development of agricultural automation and precision development. YOLOv5x was released in 2020, which greatly improved the accuracy metrics, achieving 50.7% AP on the MSCOCO dataset. You Only Look Once X (YOLOX) proposes an anchor-free structure, header separation of multitasking decoupled heads for separation, and optimized label assignment to achieve an efficient balance between real-time and reliability (Ge, 2021). The CornerNet approach converts the detection box into the coordinates of its upper-left and lower-right corners for precise localization (Law & Deng, 2018). In contrast, CenterNet achieves detection by predicting the center point along with width and height parameters (Duan et al., 2019). Fully Convolutional One-Stage Object Detection (FCOS), on the other hand, utilizes the distance from the center point to define bounding box predictions, aiding in multi-scale prediction through feature pyramid network (FPN) (Tian, Shen & Chen, 2020). YOLOX reverts to a simplified training and decoding process, drawing inspiration from these anchor-free detectors. YOLOX separates the detection heads into Decoupling Head so that each detection head performs a specific task and the learned feature information is not exactly the same (Ge, 2021). The decoupling head improves AP by 1.1% and accelerates model convergence. YOLOv6 was published in ArXiv in 2022, which designs the backbone of the model with part of the EfficientRep module and an efficient decoupling header (Li, Li & Jiang, 2022). YOLOv7, published on ArXiv in July 2022, proposes some architectural changes (extending the Efficient Layer Aggregation Network) and a series of free packages that improve accuracy without compromising inference speed, but affect training time (Wang, Bochkovskiy & Liao, 2023). The YOLOv8 model boasts anchor-free detection, reducing the number of target box predictions and speeding up non-extremely large value suppression. Its C2F module is also designed for faster model training. YOLOv8x achieves a remarkable 53.9% AP on MSCOCO. The YOLO family of detection models is dedicated to high-performance algorithms that deliver accurate results in real-time. YOLOv4 and subsequent models have focused on improving network backbones, data augmentation, and training strategies, resulting in significant accuracy improvements while maintaining real-time performance (Bochkovskiy, Wang & Liao, 2020). However, the YOLO detector still relies on non-maximum suppression (NMS) post-processing to filter out redundant bounding boxes for accurate detection. While this improves overall detection quality, it can cause a delay in detection speed. The YH-RTYO model used in this paper uses Transformer’s query instead of anchor, which does not use the filtering stage of non-extremely large value suppression, reduces the inference time, and realizes the construction of the relationship between the image target and the long-distance by using the Transformer architecture, and compares with the same scale of the YOLOv8-L model, the model mainly reduces the burden on the model, and all the other indexes are also reduced. Meanwhile, compared with the YOLOv8-L model of the same scale, the performance of the proposed model is comprehensively improved.

Transformer network

The Transformer network has been a popular topic since the emergence of vision transformer (VIT) (Han et al., 2022) in computer vision (CV). The DETR series of models, considered the representative of end-to-end target detection algorithms based on the Transformer, have continually challenged the end-to-end convolutional modeling of target detection algorithms led by YOLO (Carion et al., 2020). DETR proposes a new approach that treats target detection as a direct set prediction problem, simplifying the detection process and effectively removing the need for many manually designed components such as NMS or anchor generation (Carion et al., 2020). Despite slower training times, DETR (Carion et al., 2020) has achieved accuracy and runtime comparable to the mature and highly optimized Faster RCNN (Girshick et al., 2014) baseline on the challenging COCO target detection dataset. Deformable DETR uses the local advantages of convolutions in combination with the modeling capabilities of Transformers to enable the model attention module to focus only on a small subset of sampling points around the reference (Zhu et al., 2020). Meanwhile, the DN-DETR proposes a new denoising training method to reduce the difficulty of binary matching and speed up the convergence of DETR model training (Li, Li & Jiang, 2022). DINO further improves the DETR model’s performance and efficiency using a contrastive denoising training method, a hybrid query selection method for anchor initialization, and a look-forward-twice scheme for box prediction (Zhang, Li & Liu, 2022). With these enhancements, DINO has significantly improved model scalability in both model and data size, achieving 51.3% AP (Zhang, Li & Liu, 2022). RT-DETR is a detector released by the PaddleDetection team of Baidu Flying Paddle. It declares that it overthrew YOLO’s domination over the field of real-time detection (Zhao, Lv & Xu, 2024). A direct translation of the paper’s title is ”RT-DETR beats the YOLO family in real-time target detection”. RT-DETR outperforms the accuracy and inference speed of models such as YOLOv7 (Wang, Bochkovskiy & Liao, 2023) and YOLOv8 on COCO datasets at the same scale. However, since TRANSFORMER requires prior knowledge training, it has a greater demand for data instance samples; otherwise, it is prone to misdetection.

Multi-scale feature fusion methods

Multi-scale feature fusion methods are widely used in target detection to recognize target objects of different scale sizes. Small targets need to make the most of the detailed information in the data due to small area they occupy and the low amount of feature information. These methods employ deep and shallow feature layers to effectively detect and recognize targets for classification and regression tasks, respectively. They can also enhance robustness to object detection of different specifications. Lin, Dollár & Girshick (2017) introduced a feature pyramid structure, which employs a top-down path and lateral connections, which combine low-resolution deep semantic information with high-resolution shallow spatial information, enhancing the information of the feature graph. This method has become a commonly used network architecture component for small-target detection tasks. Liu, Qi & Qin (2018) proposed the path aggregation network (PaNet) method, which improved the effect of small-target detection. In PaFPN, a bottom-up path is followed to repair the feature loss of deep semantic information. Liu, Huang & Wang (2019) proposed Adaptive Spatial Feature Fusion, which idea adaptively learns the spatial weights of the feature map fusion at each scale. This approach improves the one-stage detector with a slight increase in computational speed and accuracy. Guo et al. (2020) proposed an augmented feature pyramid network (AugFPN) feature pyramid algorithm to enhance the FPN in response to the shortcomings. The algorithm proposes consistent supervision, which reduces the semantic differences between features at different scales prior to feature fusion by enforcing the same supervisory signal for multiple supervisory signals. In the process of feature fusion, the contextual semantic information of the stable ratio is extracted using residual feature enhancement to minimize the information loss of feature maps at the highest pyramid level. Jin et al. (2022) explained the effectiveness of the FPN and will correct the backpropagation paths between the objective function and the shallow layers of the backbone network. However, these feature pyramid structures increase the model complexity, slow down the inference, and cause the loss of deep semantic information in the feature fusion part. Yang, Lei & Zhu (2023) improved the structure of FPN and proposed an asymptotic feature pyramid network (AFPN) progressive feature pyramid to support direct interaction between non-adjacent feature layers, but due to the increased propagation routes during the training process of gradient backpropagation, the number of model parameters will also increase significantly. Zhang et al. (2024) proposed the feature pyramid network of HR-FPN, which uses a progressive method to fuse features of different scales. However, this method will greatly increase the number of parameters of the model, which is not suitable for deploying to the UAV equipment for real-time tasks. The LFPN used in this paper is optimized in the up-sampling and branch processing part to solve the problem of fuzzy feature information caused by bilinear interpolation sampling as well as the enhancement of the target features in the original feature maps so that the multi-scale features can get more adequate semantic information and more accurate spatial information.

Loss function

The loss function serves to gauge the level of how well the model fits the training data. It is a crucial component in the training part, as the model employs backpropagation to adjust the parameters and ultimately minimize the loss function, thus achieving the desired learning outcome. In the realm of target detection, popular loss functions include:

(i) Regression loss function

The mean square error (MSE) function is one of the foundations of the loss function as in Eq. (1), the principle is to measure the Euclidean distance between the predicted bounding box position and the true position, it is not suitable for classification tasks because the gradient generally tends to zero out during training. The resulting classification is incapable of determining whether the detected outcome is closer to correct or incorrect. (1) MSE=1n×∑yi−y ˆi.

The L1 loss function pass is shown as in Eq. (2), which is suitable for use in the presence of outliers, with a greater penalty for samples with large errors. This can effectively reduce the impact of outlier points on the overall results, and is commonly used in regression tasks with good robustness. (2) L1= ∑y−y ˆi

where yi and y ˆ represent the true, predicted values, respectively.

(ii) Classification loss function

When dealing with classification tasks, cross entropy loss serves as a tool to gauge the discrepancy between the anticipated probability of a target category and its real label. This is paired with softmax, the process is as in Eq. (3), where p represents the predicted probability and q represents the actual probability. (3) Hp,q=−∑xpxlogqx.

The binary cross entropy (BCE) loss function is used to detect the presence of a target as in Eq. (4), where yi is the true label value and σ(xi) is the predicted probability. (4) LBCE=−∑i=1Nyi lnσxi+1−yi ln1−σxi

(iii) Multi-task loss function for target detection

The process of target detection involves two primary tasks: classification identification and regression localization. To calculate the total loss function, the regression and classification losses are combined, with varying weights assigned to each before multiplying and adding them up. The IoU loss measures the overlap between the predicted and actual bounding boxes, and weights can be applied to adjust the loss on specific scales to meet the target detection needs. Regularization is often utilized within the loss function to ensure control over loss generation, which prevents any potential gradient vanishing or explosion. The use of generative adversarial networks can also enhance target detection by improving the quality of generated bounding boxes through the loss function between the discriminator and generator. Different loss function types can be combined and adapted to various requirements and model architectures to achieve optimal target detection performance. Ultimately, picking the right loss function is vital to the success of the task. The SIMIoU loss function proposed in this work introduces a novel approach specifically designed to address the challenges in UAV-based crop detection. Unlike traditional loss functions such as GIoU and CIoU, SIMIoU integrates factors from Inner-IoU, MPDIoU, and CIoU, and incorporates a power factor θ to enhance the penalty for samples with significant errors. This approach effectively tackles the issues of unbalanced class distribution and large-scale variations within our dataset, resulting in a significant improvement in model performance compared to existing methods. Therefore, the design of SIMIoU represents not only an extension of current research on loss function design in object detection but also a significant advancement in this field.

Object detection in UAV airborne images

Deep learning-based object detection methods for UAV remote sensing images are highly effective and efficient in both military and civilian applications. However, challenges such as unique viewpoints, complex backgrounds, and scale and orientation diversity arise due to the limitations of small UAV platforms and imaging conditions. Technological advancements have led to innovative methods to address these issues, including improvements in image data processing and handling diverse target scales and orientations. Xu, Gao & Yan (2023) proposed the multi-layer pyramid crop classification network (MP-Net) for UAV-based crop classification, addressing feature loss with a pyramid pooling module and retaining upper-level features through an information concatenation module. Luo, Li & Li (2023) proposed the evolutionary shadow correction network (ESCNet) to address shadow-induced radiometric information loss in UAV-based high-resolution remote sensing images by directly correcting shadows without requiring shadow detection. Huang, Ren & Wu (2024) proposed a feature guided enhancement module and a scale-aware weighted loss function to improve small-scale object detection in low-altitude UAV applications by refining feature discrimination and adjusting loss weights for different scales. The YH-RTYO used in this paper enhances model accuracy by addressing the limitations of RT-DETR in detecting small targets. It combines the strengths of convolutional models for localized regions, optimizing multi-scale fusion with lightweight operators and feature weighting, and adds a penalty factor to the regression loss to improve convergence and generalization.

Approach

YOLO series models are among the most prevalent one-stage detection algorithms, optimized to achieve a balance between detection accuracy and inference speed. In contrast, RT-DETR, a target detection framework based on Transformer architecture, outperforms YOLO models of similar classes in both accuracy and inference speed. RT-DETR offers superior compatibility and accuracy for handling complex scenarios, large datasets, and multi-source data. However, it incurs longer training times compared to YOLO and demands a large volume of data samples to effectively develop a priori knowledge. This requirement can lead to misdetections if data is insufficient, limiting the benefits of training with smaller datasets. As depicted in Fig. 1, this paper leverages the strengths of YOLOv8 and RT-DETR-L to develop the RTYO model, which is further refined to create the YH-RTYO model.

Figure 1 Overview of the YH-RTYO model.

Combining the advantages of YOLOv8 and RT-DETR-L, a new architecture RTYO model is constructed, and it is improved to form the YH-RTYO model. The overall model is divided into three parts: backbone, neck and head, and the loss function uses RT-DETR’s loss function. The backbone is used to extract information features in the image, and the detailed description is expressed in Fig. 2. In this part, the CFB module is optimized and designed to improve the C2F of the second and sixth layers of the model architecture. The CFB module will be explained in detail later and the structure is displayed in Fig. 4. The specific details of the neck part and the head part can be observed from Fig. 3. They use a layer of encoder layer and multi-scale fusion method LFPN for feature processing and fusion, and finally pass through a 6-layer series decoder layer for classification and position prediction.

The images are resized to a maximum of 640 ×640 after preprocessing and then fed into the feature extraction backbone, as illustrated in Fig. 2. Our backbone comprises convolutional modules and their variants, as well as a multi-scale spatial pooling layer, which together form the core architecture for target detection. This design is focused on efficient feature extraction and aims to enhance learning capabilities and network robustness. The model’s neck section is responsible for operations such as feature enhancement and multi-scale feature fusion. Various FPNs and improved modules are typically utilized in this section to boost accuracy, especially in small object detection scenarios. Figure 3 demonstrates that YH-RTYO utilizes the encoder component of RT-DETR to effectively combine semantic and location information, enabling the transfer of feature information across different scales and improving the network’s multi-scale prediction capability. YH-RTYO uses a decoder as the detection head, which contains six decoder modules, and uses 2D sen-cosine position coding. The multi-scale feature vectors processed by the Neck are input into the detection heads of different sizes. This ultimately yields bounding box and classification results. In the feature extraction portion of the YH-RTYO model, the CFB module is formed by enhancing the two-layer and six-layer C2F using the DBB concept, which enhances the model’s ability to extract image features. The LFPN architecture is designed to replace bilinear interpolation sampling with a lightweight algorithm, and the branch structure is strengthened through feature enhancement. SIMIoU, which has better convergence speed and stationarity, is used in place of GIoU in the baseline model based on the target characteristics of the data.

Figure 2 Structure of backbone in the feature extraction part of the model.

It consists of a convolutional layer, a C2F module, a CFB module, where the right is the structure of C2F, and the CFB module is clarified in Fig. 4.

Figure 3 Structure of AIFI and head in model feature processing section.

The AIFI on the left is a layer encoder, which uses 2D sine and cosine function position coding. The decoder and head on the right contains six decoder structures and finally performs classification and positioning tasks.

CFB module

The design of the feature extraction part is an important prerequisite for the good or bad performance of the model detection, the C2F of the YOLOv8 model has been greatly improved in accelerating the training speed, but it is not used to study the extraction ability. To allow the model to acquire as much helpful information as possible, it is important to reduce the impact of losing some information during feature processing and also improve the imbalance of data samples. The Diverse Branch Block enhances feature space diversity by combining branches of different scales and complexity, improving inference without sacrificing time. The C2F module features two convolutional layers arranged in a trapezoidal shape, similar to a bottle’s neck, with respect to the number of channels. Taking advantage of the fast inference speed of C2F and combining with the concept of DBB, these two convolutions are improved to improve the generalization ability of extracted features while considering the training cost. As is shown in Fig. 4, by combining four branches with different scales and complexities, including the use of Conv(1 ×1) and normalization layer, 1 ×1 convolution and 3 ×3 convolution in tandem with BatchNorm’s convolution sequence, and Average pooling operation, the diversity of feature space is enriched, achieving enhancement of a single 3 ×3 convolution. The 2-layer and 6-layer C2F modules are improved to form the CFB module, ultimately enhancing the model’s feature extraction ability.

Figure 4 CSPDarknet53 to 2-stage FPN with diverse branch block (CFB) module structure diagram.

It is a multi-branch convolutional network, which aims to enhance the robustness of feature learning.

Feature pyramid LFPN module

The FPN method is a highly efficient and versatile technique for enhancing model performance in target detection. In this study, we examine the application of FPN for feature extraction within the feature pyramid module. Due to the repeated use of up-sampling operations and the reliance on simple 1 × 1 convolutional branches for feature processing, issues such as feature redundancy and errors can arise, particularly leading to increased background noise interference. This interference can impair the model’s ability to accurately predict target boundaries. Therefore, employing more efficient up-sampling methods and advanced branching techniques prior to feature fusion is crucial.

The baseline RTYO model utilizes the conventional bilinear upsampling method, which is uniform and lacks consideration of the semantic information of the feature map. Alternative methods such as deconvolution and dynamic filters use varying upsampling kernels at each feature map position, but these approaches are computationally expensive and not suitable for practical applications. In contrast, this study employs CARAFE, a lightweight upsampling operation, as illustrated in Fig. 5. CARAFE predicts a distinct upsampling kernel for each position based on the input feature map and reorganizes the features accordingly.

Figure 5 Carafe module structure diagram.

It is an optimization algorithm for lightweight sampling of upsampling.

Analysis of the branch structure revealed that objects in the original image feature layer were underutilized. As sampling progressed and layers deepened, features became increasingly blurred. To address this, we transformed the 1 ×1 convolution layer into a feature enhancement module (FEM), as shown in Fig. 6. The FEM pools feature vectors along the width and height dimensions to evaluate the importance of each channel using a feature map (fm). It then adjusts channel values by multiplying the feature vectors with the weights derived from fm and summing the original channel values via residual connections. Finally, a 1 ×1 convolution is used to screen the channels, resulting in adjusted feature vectors.

Figure 6 FSM feature enhancement module diagram.

This multi-scale fusion gives full play to the role of features from the branch part, and uses weighted attention to strengthen the features, so as to reduce the fuzzy information in the fusion of different scales.

Figure 7 illustrates the optimization of the LFPN module, which includes two key improvements: the CARAFE upsampling method introduced earlier and the FSM method, which enhances the differentiation of target and background features. The LFPN module improves the model’s ability to learn features from both positive and negative samples by reducing sampling loss and reinforcing target features.

Figure 7 Structure of LFPN multi-scale feature fusion module.

The features on the branch are processed by feature enhancement and then input to the connection. The feature information is sampled by CARAFE for learning, and the features on the branch are enhanced by FSM to get the enhanced feature vector, and the two are fused to form a multi-scale fused feature vector with richer semantic information and complete spatial information retention.

SIMIoU loss function

A good loss function is a crucial element in the learning process of the model. It quantifies the error in the training phase by comparing the predicted and actual values of sample data. Through backpropagation, the model updates its parameters in the direction of the real value to minimize the loss value. A smaller loss value indicates that the model’s predicted information is closer to the actual information in the sample data, making the model more robust, we use the FocalLoss classification loss function as in Eq. (5), which prioritizes the loss of rare categories to improve the model’s accuracy. (5) FocalLoss=−αt1−ptγ× logpt.

The variable αt is utilized to address the issue of imbalanced positive and negative samples in the dataset, while γ is utilized to address the issue of imbalanced difficult and easy samples. The bounding box regression loss function of this model employs the GIoU loss function of RT-DETR as in Eq. (7). (6) IoU=P∩GP∪G

(7) GIoU=IoU−Union−P∩GUnion.

The efficient intersection over union (EIOU) loss function is a widely used approach that takes into account multiple influencing factors when dealing with small targets, as in Eq. (8). The EIoU loss can effectively use the actual aspect ratio to achieve faster convergence and more accurate localization results. (8) LEIoU=1−IoU+ρ2b,bgtc2+ρ2w,wgtcw2+ρ2h,hgtch2.

SIoU is a widely used loss algorithm that measures the quality of the prediction box from an angle, which is divided into distance loss and shape loss. Angular loss is the base building block of SIoU, and the equation for the angular loss Λ is shown as in Eqs. (9), (10) and (11) are complements of Eq. (9). (9) Λ=1−2× sin2arcsinchσ−π4

(10) chσ= sinα

(11) σ=bcxgt−bcx2+bcygt−bcy2

where bcxgt bcygt are the coordinates of the center of the true frame and bcx bcy are the coordinates of the center of the predicted frame. (12) ch= maxbcygt,b cy− minbcygt,b cy.

The Δ equation is as in Eqs. (13) and (14). (13) Δ= ∑t=x,y1−e−γρt=2−e−γρx−e−γρy

(14) ρx=bcxgt−b cxcw2,ρ y=bcygt−b ych2,γ=2−Λ.

The shape loss equation is as in Eqs. (15) and (16). (15) Ω=1−e−wwθ+1−e−whθ

(16) ww=w−wgt maxw,wgt,wh=h−hgt maxh,hgt.

The SIoU loss function is shown as in Eq. (17). (17) Losssiou=1−IoU+Δ+Ω2.

In this paper, we introduce a loss function specifically designed for the current task scenario, termed SIMIoU, as defined in Eq. (18). This loss function takes into account both the common area and the minimum enclosing frame of the predicted and ground truth boxes, as detailed in Eq. (19) under Inner-IoU. To address substantial deviations more effectively, we incorporate a parameter θ that intensifies penalties for samples with significant errors. Additionally, we integrate considerations from the MPDIoU approach regarding diagonal distance factors and the CIoU approach concerning aspect ratio factors. (18) SIMIoU=inneriouθ−d1h2+w2+d2hwθ−12×α×γ.

In this context, d1 refers to the square of the distance between the upper left corners of the prediction and actual boxes, while d2 denotes the square of the distance between their lower right corners. Additionally, h and w correspond to the height and width of the image, respectively. Following normalization, both h and w equal one. (19) inneriou=interunion.

In object detection, as in Eq. (19), intersection over union (IoU) is a commonly used metric to evaluate the accuracy of the predicted bounding box. Inter is the region where the expanded or contracted predicted box overlaps the actual expanded or contracted box, and union is the smallest region that contains the two resized bounding boxes. The IoU value is obtained by dividing the intersection area by the union area, which ranges from 0 to 1. A higher IoU value indicates a good match between the predicted and actual bounding boxes. (20) γ=4π2×arctanwgthgt− arctanwh2

(21) α=γ1−IoU+γ.

Consider the aspect ratio of the prediction frame, as in Eq. (20), γ is a parameter that measures the consistency of the aspect ratio, and as in Eq. (21), α is the coefficient that balances γ. The overall loss function is shown in Eq. (22). (22) LossSIMIoU=1−SIM=1−inneriouθ+d12+d22θ+12×α×γ.

Experiment

Experimental environment and setup

This paper presents experimental verification conducted on the NVIDIA A40, with all models run on the PyTorch version 1.13.1 environment. The random number seed is set to 0, the learning rate is set to 0.0001, weight decay to 0.0001, warm-up iteration to 2000, total epochs to 300 rounds, and batch size to 4. For training and validation, the oilpalmuav dataset samples were resized to 640 ×640 pixels.

Experimental dataset

UAVs equipped with cameras have found rapid deployment in various agricultural applications in the real world. These applications include field fire prevention, disease inspection, insect monitoring, grass pest control, and growth status detection. This paper uses the oilpalmuav dataset (Zheng et al., 2021) to learn and validate a model for detecting the growth status of crop oilpalm in UAV scenes. The dataset comprises 3,999 training images captured by UAV and 500 high-quality images for the validation dataset. The dataset contains five classes: health oil palm, dead oil palm, mismanaged oil palm (grass), yellow oil palm, and small oil palm. The dataset covers different shooting times, multiple local areas, and varying light and shade changes. The oilpalmuav dataset is rich in instance samples, as shown in Table 1 and Fig. 8, which displays the detailed instance distribution in the dataset.

Table 1 The instances distribution table for the oilpalmuav dataset.

The dataset comprises 3,999 training images captured by UAV and 500 high-quality images for the validation dataset. The dataset contains five classes: health oil palm, dead oil palm, mismanaged oil palm (grass), yellow oil palm, and small oil palm.

Categories	Oilpalm	Healthy	Dead	Dismanaged (grass)	Yellow	Small	
train_instances	262811	201303	599	953	3103	56893	
val_instances	32846	24884	61	95	411	7395	

Figure 8 Oilpalm crop growth status category.

Healthy, Dead, Grass, Small, Yellow.

Evaluation metrics

To thoroughly assess the YH-RTYO model’s performance in crop growth conditions, this paper employs a range of evaluation measures, including precision in Eq. (23), recall in Eq. (24), AP50 in Eq. (25), mAP in Eq. (26), Para, and FLOPS. (23) Precision=TPTP+FP

(24) Recall=TPTP+FN.

In object detection, true positives (TP) are correctly identified target boxes, whereas false positives (FP) are incorrectly identified boxes. False negatives (FN) represent missed target boxes, calculated as the difference between the total number of target boxes and TP. Precision and recall for each class are evaluated using a precision-recall (P-R) curve, and the mean average precision (mAP) is determined by the area under this curve. The mAP value is typically calculated over a range from 0.5 to 0.95 in steps of 0.05. The formula for this calculation is given in Eq. (26), where m represents the number of target classes. (25) AP= ∫01prdr

(26) mAP=1m∑1mAP.

Ablation experiments

The ablation experiments were conducted on the oilpalmuav dataset mentioned earlier in this section. In this paper, the latest official YOLO model YOLOv8 and RT-DETR are combined to construct a new model RTYO, the main architecture is modified in RT-DETR, the configuration file uses the RT-DETR default configuration file, and the backbone and scale use YOLOv8′s settings and pre-training files. The Innovative modules were respectively designed to enhance the model’s performance on the dataset. Firstly, during training, the CFB module utilizes multi-branch convolutional combinations to bolster the model’s robustness, while during testing, it reverts to a single-branch structure. Additionally, the feature pyramid LFPN focuses on selecting adjacent scale features to fortify features and mitigate feature loss caused by concatenation. Finally, the training process incorporates the SIMIoU loss function, which comprehensively considers factors such as angles and multiple diagonal distances between two target boxes. RTYO serves as the benchmark algorithm to observe the actual impact of each module’s improvement, thereby validating the effectiveness of these model enhancements. The specific ablation experiments are shown in Table 2 and the final RTYO method has significant improvement in precision, recall, AP50, and mAP evaluation indexes compared with RT-DETR-Resnet50, which combines the advantages of each one and balances the precision and the number of parameters for RT-DETR-L and YOLOv8.

Table 2 Table of ablation experiments for the YH-RTYO model based on the oilpalmuav dataset.

On images with width and height of 640, the proposed method has obvious superior effects in terms of precision, recall, AP50, mAP, Para, and FLOPS.

Method	Size	Precision
(%)	Recall
(%)	AP50
(%)	mAP
(%)	Para
(M)	FLOPS
(G)	
RTYO	640	94.6	92.6	96.2	71.7	37.67	142.9	
RTYO+SIMIoU	640	95.7	92.5	96	72.4	37.67	142.9	
RTYO+CFB	640	93.9	95.6	96.3	74.3	37.67	142.9	
RTYO+LFPN	640	94.9	95.5	97.1	75	38.14	143.2	
RTYO+CFB+LFPN	640	95	95.1	96.9	75.5	38.14	143.2	
YH-RTYO(ours)	640	95.9	94.9	97	75.67	38.14	143.2	

As shown in Table 2 under the same conditions, despite a relative increase in both the parameter count and computational workload, RTYO model surpasses the RT-DETR-L model with a 3.2% enhancement in recall rate and an 8% improvement in mAP. In the multi-scale fusion part of the improved LFPN, with more efficient CARAFE up-sampling and FSM enhancement branch, in the case of a slight increase in the number of parameters, all the indicators have been significantly improved, including a 3.3% improvement in the small target indicator mAP, recall increased by 2.9%, indicating that multi-scale fusion of the information between the larger impact on the detection of small targets. By leveraging C2F’s rapid inference speed and incorporating the DBB concept, we enhanced the richness of the feature space by merging branches with varying scales and levels of complexity to optimize single convolutions. Our creation of the CFB module was validated by ablation experiments, which demonstrated a 3.0% increase in recall and a 2.6% improvement in mAP. These results confirm that the proposed model has a clear advantage in feature extraction to reduce feature loss while improving the overall performance.

At the same time, as shown in and Tables 3 and 4, the effect data of RTYO model and YH-RTYO model in different categories are displayed. The experiment shows that the model detection effect under five categories is significantly improved, and the Dead type is significantly improved. According to the characteristics of the dataset and the overall structure of the model, the SIMIoU loss function is shown in the table, which improves the detection effect significantly in the evaluation metrics, and there is a 0.7% enhancement in the mAP index, as well as as as shown in Table 5 and Fig. 9, and according to the comparison with the existing popular several loss functions, SIMIoU is also faster and more stable in the training process.

Table 3 Detection effect of different kinds of instances in RTYO model.

Category	Method	Precision (%)	Recall (%)	AP50 (%)	mAP (%)	
Dead	RTYO	93.5	94.3	98	77.3	
Healthy	RTYO	97.3	96.4	98.2	82.9	
Grass	RTYO	91.4	89.1	93.3	56.9	
Small	RTYO	94.7	88.1	93.8	64.3	
Yellow	RTYO	96.1	95	97.7	76.9	

Table 4 Detection effect of different kinds of instances in YH-RTYO model.

Category	Method	Precision (%)	Recall (%)	AP50 (%)	mAP (%)	
Dead	YH-RTYO	97.9	98.4	98.6	85.5	
Healthy	YH-RTYO	97.2	96.9	98.4	84.8	
Grass	YH-RTYO	91	95.8	94.8	62.2	
Small	YH-RTYO	94.7	90.5	94.8	67.6	
Yellow	YH-RTYO	95.7	96.6	99.2	79.4	

Table 5 Comparative experiments of loss function of RTYO model on oilpalmuav dataset.

The better the loss function, the faster the model converges during training, and the better the model is. The loss function in this paper has a good performance in the dataset of this research task.

IoU	Precision (%)	Recall (%)	AP50 (%)	mAP (%)	Para (M)	FLOPS (G)	
GIoU	94.6	92.6	96.2	71.7	37.67	142.9	
EIoU (Zhang, Ren & Zhang, 2022)	95.3	93.4	96.2	69.8	37.67	142.9	
Siou (Gevorgyan, 2022)	95.3	93.0	96.5	71.2	37.67	142.9	
SIMIoU	95.7	92.5	96.0	72.4	37.67	142.9	

Figure 9 Comparison experiment of accuracy metrics between YH-RTYO algorithm model and other models on oilpalmuav dataset.

In the figure, the loss function SIMIoU (red) of this paper is prominent in the performance of stationary and loss values.

In a word, LFPN focuses on the feature processing of local branch positions of multi-scale fusion, strengthens the distribution position of features, and constructs a C2F module to enhance the ability of feature extraction of part of the learning information. Under a more appropriate loss function, the aspect ratio and angle factor are considered, the model’s accuracy increased by 1.1%, and its average precision improved by 0.7%.

Contrasting experiments

In order to assess the efficacy of the algorithm presented in this paper for detecting crop growth in UAV images, a comparison and analysis was conducted against current state-of-the-art target detection algorithms. The evaluation index of various algorithms was used to confirm the superiority of this paper’s improved model. Table 6 compares various target detection algorithms on the oilpalmuav validation set based on the experimental metrics discussed in this paper. The results indicate that the proposed algorithm outperforms all other classical network models in these metrics while maintaining an effective balance between the number of parameters and computational load. Notably, the mAP value of the proposed algorithm significantly exceeds that of other classical models. As shown in Tables 6, 7, and Fig. 10, YOLOv8 represents the latest advancement in the YOLO series, while RT-DETR is a notable achievement from the DETR series presented at CVPR.When compared on the same scale, the improved model presented in this paper outperforms YOLOv8, with an increase in accuracy by 2.7%, recall by 0.3%, AP50 by 11.4%, and mAP by 0.57%, while simultaneously reducing the number of parameters by 12.5% and the amount of computation by 13%. RT-DETR-R50 and RT-DETR-L are models at the same scale as YOLOv8-L. RT-RTYO has fewer parameters than RT-DETR-R50 and more parameters than RT-DETR-L. Nonetheless, when compared to the RT-DETR model at the same level, the improved model presented in this paper outperforms RT-DETR-L, with an increase in accuracy by 1.7%, recall by 5.5%, AP50 by 2.4%, and mAP by 11.93%. The experimental comparison of multiple algorithms indicates that the proposed target detection algorithm in this paper shows a significant improvement in the accuracy of detecting crop growth and health in UAV aerial images compared to standalone YOLOv8 or RT-DETR algorithms.

Table 6 Comparison experiment of accuracy metrics between YH-RTYO algorithm model and other models on oilpalmuav dataset.

The YH-RTYO model was compared with other models, some of which used different frame indicators, so two of them selected the same indicator for filling in and different indicators for blank processing.

Method	Precision (%)	Recall (%)	AP50 (%)	mAP (%)	
YOLOv3 (Redmon & Farhadi, 2018)	94.6	73.1	83.7	73.9	
YOLOv5 (Kim et al., 2022)	91.6	74.6	83	71.6	
YOLOv7 (Wang, Bochkovskiy & Liao, 2023)	76.7	68.3	73.5	53.1	
YOLOv8 (Jocher, Chaurasia & Qiu, 2023)	93.2	74.6	85.6	75.1	
RT-DETR-R50 (Lv W et al.,2023)	86.8	82.1	87.4	50.7	
RT-DETR-L (Lv W et al.,2023)	94.2	89.4	94.6	63.7	
RetinaNet (Lin, Goyal & Girshick, 2017)	–	–	66.6	62.9	
Fcos (Tian Z et al., 2019)	–	–	61.1	55.5	
RTYO (ours)	94.6	92.6	96.2	71.7	
YH-RTYO (ours)	95.9	94.9	97	75.67	

Table 7 Experimental table comparing the size of the YH-RTYO model with other models on the oilpalmuav dataset.

The YH-RTYO model and other models are considered in terms of parameters and computational complexity to ensure that the model balances performance and size.

Method	Size	Para (M)	FLOPS (G)	
YOLOv3 (Redmon & Farhadi, 2018)	640 ×640	103.66	282.2	
YOLOv5 (Kim et al., 2022)	640 ×640	53.13	134.7	
YOLOv7 (Wang, Bochkovskiy & Liao, 2023)	640 ×640	36.50	103.2	
YOLOv8 (Jocher, Chaurasia & Qiu, 2023)	640 ×640	43.61	164.8	
RT-DETR-R50 (Lv W et al.,2023)	640 ×640	41.96	129.6	
RT-DETR-L (Lv W et al.,2023)	640 ×640	31.99	103.5	
RetinaNet (Lin, Dollár & Girshick, 2017)	640 ×640	36.41	82.27	
Fcos (Tian Z et al., 2019)	640 ×640	32.12	78	
RTYO (ours)	640 ×640	37.67	142.9	
YH-RTYO (ours)	640 ×640	38.14	143.2	

Figure 10 Plot of accuracy metrics for RTYO, YOLOv8-L, RT-DETR-, YH-RTYO.

In the figure, the proposed method can reach a higher position and converge faster than other methods.

A confusion matrix is a visual representation that shows the performance of a model through tables and numerical values. In the matrix, each row represents the actual category, and each column represents the predicted category. By using the confusion matrix, you can see the proportion of correct detections, missed detections, and wrong detections in the model’s results. To evaluate the model’s performance, the confusion matrix is used to demonstrate the improved performance of the model, as shown in Fig. 11. The four plots in the figure represent the confusion matrices of RTYO, YH-RTYO, YOLOv8-L, and RT-DETR-L, respectively. YH-RTYO Fig. 11B), the optimized model, has improved values on the diagonal line compared to the original model (Fig. 11A), YOLOv8-L (Fig. 11C), and RT-DETR-L (Fig. 11D). This proves that the detection effect is highly accurate, with less error and more balanced prediction of different categories.

Figure 11 (A–D) Confusion matrix for RTYO, YOLOv8-L, RT-DETR-L, and YH-RTYO.

Larger values on the diagonal of the confusion matrix indicate better model performance.

Feature visualization

To evaluate the RTYO algorithm’s performance in real-world scenarios, UAV aerial images from diverse and complex scenes within the oilpalmuav validation set were used. The results of these detections are shown in Fig. 12. The left side shows the baseline model, the right side shows the improved model, the box contains instances of oil palm. In the three comparison examples in Figs. 12A, 12B and 12C, comparing each row, the place surrounded by the red circle is the comparison of the detection effect of the two models, and the missing detection and false detection occur on the left. Especially in the edge, the incomplete display of the overall representation of the plant is not identified, and the accuracy and recall rate of the right picture have been greatly improved, and the overall visualization effect is more obvious. The improved model enhances the detection accuracy and positioning is more accurate.

Figure 12 Comparison of RTYO and improved YH-RTYO detection in oilpalmuav dataset.

Rows (A), (B), and (C) show the comparison plots where the validation set contains different categories and locations, and the difference in model detection effect can be clearly found from the places marked in red on the left and right sides.

Overall, the improved algorithm has higher detection accuracy, while reducing the phenomenon of missed detection and false detection. Because the densely distributed regional targets can accurately identify the exact location of each target, but also exclude the influence of interfering objects, such as soil, stones, etc., to correctly classify and locate the target, this paper’s method in different lighting conditions, shooting angle, distribution of the actual scene have shown better detection effect, can meet the needs of UAV aerial images of crop growth and health status detection tasks.

Validation of other datasets

The Rectangular Frame-based Rape Flower Cluster Dataset (RFRB) is a specialized UAV image dataset designed for object detection during the flowering stage of rapeseed plants (Li, Wang & Qiao, 2023). It includes high-resolution RGB images that capture various field conditions and growth stages. Detailed rectangular annotations are provided, facilitating the training and evaluation of deep learning models for counting and detection tasks. To validate the generalizability of the proposed model for detecting small crop targets in UAV scenarios, comparative experiments were conducted using the RFRB dataset. As is shown in Table 8, a detector’s performance cannot be assessed by accuracy alone. Although other mainstream models achieve high accuracy, their recall rates fall below 73%, meaning over 27% of objects are not detected. In contrast, the proposed model not only maintains a high accuracy but also enhances generalization through the CFB and LFPN modules, improving recall by 7.4% and increasing AP50 by 3.8%.These results indicate that the proposed method also achieves similar improvements on the RFRB dataset. As illustrated Fig. 13A YOLOv8 exhibits notably high recognition scores but suffers from significant instances of missing rapeseed flowers. Figure 13B for RT-DETR-R50 displays some missed detections within the blue bounding boxes and generally lower object recognition scores. In contrast, the proposed method, shown in Fig. 13C, demonstrates pronounced recognition effectiveness, characterized by higher object detection scores and fewer instances of omission compared to the aforementioned approaches.

Table 8 Comparison experiment of accuracy metrics between YH-RTYO algorithm model and other models on RFRB dataset.

The effectiveness of the proposed method can be verified by conducting experiments on other dataset.

Method	Precision (%)	Recall (%)	AP50 (%)	mAP (%)	
YOLOv3 (Redmon & Farhadi, 2018)	0.917	0.721	0.839	0.487	
YOLOv5 (Kim et al., 2022)	0.923	0.721	0.841	0.498	
YOLOv8 (Jocher, Chaurasia & Qiu, 2023)	0.919	0.726	0.844	0.518	
RT-DETR-R50 (Zhao, Lv & Xu, 2024)	0.708	0.523	0.61	0.166	
Fcos (Tian, Shen & Chen, 2020)	–	–	0.258	0.115	
YH-RTYO (ours)	0.906	0.80	0.882	0.545	

Figure 13 (A–C) Comparison plot of the detection performance of different methods on the RFRB dataset.

By comparing the effect of the model with other mainstream methods on the RFRB data set, the AP index is higher in the case of less parameters and calculation, which proves that the method designed in this paper has certain efficiency and superiority.

Conclusions

This paper addresses the challenges of identifying different crop states in drone aerial images, including low detection accuracy and high miss rates. Given the limited research and datasets available for crop growth state detection, and the need for models with strong real-time performance and appropriate parameters, we introduce the YH-RTYO detector. This model combines the advantages of both Transformer and YOLO convolutional architectures. Compared to YOLOv8, YH-RTYO demonstrates superior performance in terms of accuracy, recall, AP50, mAP, parameter count, and computational load. Although YH-RTYO has more parameters than RT-DETR, it still has significantly fewer parameters compared to YOLOv8. The model shows substantial improvements over RT-DETR-L in accuracy, recall, and AP50, with a particularly notable enhancement in mAP, addressing the issues of slow inference speed and large model sizes associated with traditional convolutional networks.

The YH-RTYO model leverages LFPN to enhance multi-scale feature fusion, effectively integrating small target feature layers with deep semantic information. The incorporation of the CFB module enables better learning of generalized features, reducing the risk of overfitting to specific tasks and improving detection capabilities. Additionally, the design of the SIMIoU loss function, tailored to the distribution characteristics of the oilpalmUAV dataset, accelerates model training convergence. These advancements significantly enhance performance in multi-scale target detection tasks, providing robust support for addressing small target detection challenges in practical applications.

Theoretically, this approach departs from traditional single convolution or Transformer methods by combining the strengths of both to construct a novel model. Empirically, the proposed algorithm delivers better detection results and higher accuracy compared to various other object detection algorithms. Practically, the method’s compact size, fast real-time performance, and high detection accuracy make it well-suited for deployment in inspection drones for precise crop growth monitoring. This meets the needs of integrating and advancing intelligent agriculture, smart devices, and smart technology applications. Future research will focus on developing methods to accurately identify crop positions and enhance the detection of crop growth states without increasing network complexity, addressing issues such as occlusion in dense areas, challenges in detecting small targets from a top-down perspective, and improving accuracy.

Additional Information and Declarations

Competing Interests

Author Contributions

Data Availability

The authors declare there are no competing interests.

Yihang Li conceived and designed the experiments, performed the experiments, performed the computation work, prepared figures and/or tables, and approved the final draft.

WenZhong Yang performed the computation work, authored or reviewed drafts of the article, and approved the final draft.

Zhifeng Lu performed the computation work, authored or reviewed drafts of the article, and approved the final draft.

Houwang Shi performed the computation work, authored or reviewed drafts of the article, and approved the final draft.

The following information was supplied regarding data availability:

The MOPAD dataset is available at Github and at Zheng et al. (2021):

- https://github.com/rs-dl/MOPAD

- https://doi.org/10.1016/j.isprsjprs.2021.01.008.

The RFRB dataset is available at Github and is described in Ji et al. (2023):

- https://github.com/CV-Wang/RapeNet.

- https://doi.org/10.1186/s13007-023-01017-x.

The code is available at Github and Zenodo:

- https://github.com/MLL768020/RTDETR-20231127/tree/master

- MLL768020, & AINISHDD. (2024). MLL768020/RTDETR-20231127: v1.0 (v1.0-yhrtyo). Zenodo. https://doi.org/10.5281/zenodo.13709938.

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
