# Peer review of "YH-RTYO: an end-to-end object detection method for crop growth anomaly detection in UAV scenarios"

_PeerJ Computer Science, doi:10.7717/peerj-cs.2477_

## Round 0.1 · original submission · Major Revisions

Dear authors,

Thank you for submitting your manuscript for review. According to the reviewers' comments, the manuscript requires Major Revisions. Please read carefully at the reviewers' comments and make revisions correspondingly. In particular, since the proposed method is not too much better than state-of-the-art algorithms, the novelty of the work should be made clear. Also, the authors may want to improve the overall structure of the paper and check carefully for typos.

All the best.

Reviewer 1 ·

Basic reporting

1. There are some grammatical errors in this paper, such as lines 39-40. In addition, there is a problrm with the semantics of " we combine the precision of YOLO series models for small targets with the efficient computation and high recall of Transformer-based DETRs", where "combining precision and recall" is an incorrect statement.

2. With the key innovation about "Multi-Scale Fusion and Model Optimization" in this paper, the following description of the method proposed in this paper lacks a description of what kind of optimization problem exists, what the optimization method is, and what advantages the optimization can bring. Please consider whether this innovation is appropriate.

3. Please consider briefly describing the background and the problem by one or two sentences in the abstract and making the conclusion more concise.

Experimental design

The experimental design of this paper is reasonable and effective.

Validity of the findings

No comment

Additional comments

Please consider restructuring this paper so that it is more logical in describing the background, problem, and proposed method.

Reviewer 2 ·

Basic reporting

1). In the "Abstract" section, only the content of the "Results" is presented, without introducing the work of the paper, and "Result" is misspelled as "zResult".
2). The formula on line 238 needs to be centered.
3). There are two consecutive periods ".." in line 274.
4). The font size in the "Validation of other datasets" section is not consistent with the rest of the document's font size.
5). Replace Figure 5 with a higher clarity image.

Experimental design

In the experiments on the RFRB dataset, the precision shown in the results does not seem to have an advantage, and the model does not appear to show significant performance improvements in other experiments either.

Validity of the findings

There is a lack of deeper analysis in the experimental results section, making it difficult to demonstrate the outstanding contribution of the paper's work.

Reviewer 3 ·

Basic reporting

1.In Related Work, it only focuses on CNN, transformer network, multi-scale feature fusion methods and the loss function. It is not enough to show that the method proposed by the author complements the existing research. The related work about object detection method in UAV Scenarios should be reviewed.
2. The English language should be improved to ensure that an international audience can clearly understand your text. For example, "End-to-end object detection method" in the title should be "An end-to-end object detection method"; in Line 199, "...response to the shortcomings0." should be "...response to the shortcomings." ; in Line 204, "... Jin et al.(2022) explains" should be "... Jin et al.(2022) explained".

Experimental design

1. In Experimental datasets, the oilpalmuav dataset is depicted without the RFRB dataset. However, authors conduct the experiments on the RFRB dataset, as show in Table 8. And the RFRB dataset should be depicted in Experimental datasets part.
2. As shown in Table 2, the best precision difference is only 2%, the best mAP difference is only 3.97%; and other metrics are not better than other method. It doesn’t seem to have efficacy advantage in the current evaluation scheme.

Validity of the findings

In this paper, the novelty is not clear. The authors only utilize the advantages of the YOLOv8 and RT-DETR-L models to propse the object detection method. Besides, previous methods for detection by using AI techniques have been proposed. The authors should show the novelty and advantages of the proposed method compared with the existing methods.

Additional comments

In this paper, authors combine the advantages of YOLOv8 and RT-DETR-L to propose the RTYO model and the improved the YH-RTYO model. The experimental results on the oilpalmuav dataset and the RFRB dataset show that the proposed models could detect the object in the UAV scenarios.

---

## Round 0.2 · accepted · Accept

Dear authors, we are pleased to verify that you meet the reviewer's valuable feedback to improve your research.

Thank you for considering PeerJ Computer Science and submitting your work.

Reviewer 1 ·

Basic reporting

no comment

Experimental design

no comment

Validity of the findings

no comment

Additional comments

no comment

Reviewer 2 ·

Basic reporting

The article is neatly formatted, with no grammatical or spelling errors, and it meets the standards for acceptance.

Experimental design

The experimental design is rational. The results are substantial and have been appropriately compared with multiple lastest algorithms, such as YOLOv8, RT-DETR-L.

Validity of the findings

The analysis of the experimental results is appropriate and has verified that the algorithm proposed in the article has a certain practical value: The model reduces the number of parameters by 13%, and the AP50 and AP values of this model exceed those of the YOLOv8 baseline by 3.8% and 2.7%, respectively.

Reviewer 3 ·

Basic reporting

Small object detection via Unmanned Aerial Vehicles (UAV) is crucial for smart agriculture, enhancing yield and efficiency. This manuscript addresses the issue of missed detections in crowded environments by developing an efficient algorithm tailored for precise, real-time small object detection. And this manuscript is well written, well organized, and thus easy to follow.

Experimental design

In this manuscript, the research question is well defined, and the proposed model is described sufficiently well. Besides, the experimental results show that the proposed YH-RTYO model outperforms others in key detection metrics.

Validity of the findings

This topic is interesting. And its novelty is enough to publication

Additional comments

The authors have answered my question very well. I have no other questions and suggest the editor accept it.